# Which Aspects of Work Safety Satisfaction Are Important to Mental Health of Healthcare Workers during COVID-19 Pandemic in Poland?

**DOI:** 10.3390/ijerph20042870

**Published:** 2023-02-06

**Authors:** Katarzyna Gustavsson, Zuzanna Goetz-Kundera, Magdalena Flaga-Łuczkiewicz, Adam Wichniak

**Affiliations:** 1Department of Clinical Neurophysiology, Institute of Psychiatry and Neurology, 02-957 Warsaw, Poland; 2Department of Psychiatry, Bielański Hospital, 01-809 Warsaw, Poland; 3Mental Health Center for Medical Doctors and Students, Dialog Therapy Centre, 02-703 Warsaw, Poland; 4Third Department of Psychiatry, Institute of Psychiatry and Neurology, 02-957 Warsaw, Poland

**Keywords:** healthcare workers, mental health, COVID-19

## Abstract

The outbreak of the COVID-19 pandemic increased the occupational burden experienced by healthcare workers. The aim of this study was to investigate a change in work satisfaction during the pandemic and specific factors contributing to mental health among healthcare providers. We obtained data from 367 healthcare professionals. Respondents were asked about their satisfaction with selected aspects of work (clarity of procedures, access to personal protective equipment, the flow of information, financial stability and general security) during the epidemic and retrospectively how satisfied they were before the outbreak. They also completed measures assessing mental health: the World Health Organization-Five Well-Being Index, the Patient Health Questionnaire-9, the Generalized Anxiety Disorder-7 scale and the Insomnia Severity Index. The results showed that satisfaction with all safety-related work aspects decreased during the pandemic. The flow of information and financial stability were significant predictors of WHO-5, PHQ-9 and ISI scores. GAD-7 scores were predicted by satisfaction with the clarity of procedures, the flow of information and financial stability. The COVID-19 pandemic significantly changed the lives of everyone. However, due to conditions of employment in Polish healthcare, the COVID-19 pandemic put a great financial strain in addition to pandemic stressors specific to medical staff.

## 1. Introduction

Working in healthcare has been consistently shown to be a high-stress profession. The outbreak of the COVID-19 pandemic increased the occupational burden even more. The rapid spread of SARS-CoV-2 placed a huge strain on healthcare systems. Especially because, during this time, there was no vaccine or therapy approved for the virus. Healthcare workers started to experience high levels of stress directly related to the pandemic [1]. There has been an urgent need for a well-functioning healthcare system that is dependent on the efficiency and health of medical staff.

The outbreak of the COVID-19 pandemic dramatically influenced the working environment in clinics and hospitals. The epidemiological situation changing daily or hourly influences one’s perception of personal risk [2]. In Poland, many wards were suddenly, without any preparation, transformed into “COVID-19 wards”, regardless of the previous specialty (also, for example, dermatologic, orthopedic or gynecologic). Physicians and nurses from those wards were expected to work with COVID-19 patients without any additional training. There was a forced rotation of personnel to the temporary COVID-19 hospitals. At the same time, patients with health conditions other than COVID-19 could not receive their standard care due to a healthcare system overload, which frustrated the healthcare workers as well. The situation was dynamic in primary care as well. General practitioners were supposed to switch to telemedicine, even though they had not practiced it before and therefore had no tools for telemedical health care. Polish healthcare workers during the first wave of the pandemic often had to improvise in order to deliver adequate care to their patients [3].

Dealing with uncertainty, tiredness and frequent changes of procedures in hospitals placed healthcare personnel at a high risk of mental health deterioration [4,5]. Meta-analyses have shown that a significant percentage of healthcare workers during the COVID-19 pandemic suffered from depression, anxiety and insomnia [6,7,8,9].

There has been a clear need to identify modifiable factors associated with psychological distress among healthcare workers. The initial shortage of personal protective equipment was a significant problem in the early stages of the pandemic. In hospitals in Greece, the lack of personal protective equipment (PPE) was one of the predictors of psychological distress among healthcare providers [5,10,11,12]. Shortage of PPE has been shown to be linked to anxiety [13] and general mental health [5,14]. In a study by Khajuria et al. [15], participants who reported not having been issued appropriate PPE or not having received adequate training were more likely to experience poor mental health.

Apart from the insufficient amount of PPE, healthcare personnel reported other issues problematic during the pandemic, such as constantly changing regulations that resulted in uncertainty about procedures, inconsistent or ineffective communication and potential salary cuts [1,16,17,18].

This study aimed to investigate a change in work safety satisfaction during the pandemic and current satisfaction with aspects of work safety as factors contributing to mental health among healthcare providers.

## 2. Materials and Methods

A survey was carried out in May 2020 during the first wave of the COVID-19 pandemic in Poland as an online anonymous volunteer-based study advertised via social media and direct messages. Respondents were asked to rate the percentage in which they were satisfied with selected aspects of work (clarity of procedures, access to personal protective equipment, the flow of information, financial stability and general security) during the epidemic and how retrospectively satisfied they were before the outbreak. They also completed measures assessing mental health: The World Health Organization-Five Well-Being Index (WHO-5 [19]) is a 5-item scale to measure their general emotional wellbeing, with raw scores ranging from 0 to 25 to be multiplied by 4 to obtain a final score of between 0 and 100; The Patient Health Questionnaire-9 (PHQ-9 [20]), with scores ranging from 0 to 27, and the Generalized Anxiety Disorder scale (GAD-7 [21]), with scores from 0 to 21, two of the most frequently used diagnostic self-report scales for screening, diagnosis and severity assessment of depression and anxiety; The Insomnia Severity Index (ISI [22]) is a widely used brief instrument screening for insomnia consisting of seven questions about sleep problems with scores ranging from 0 to 28.

Following the regulations of the Bioethical Committee of the Medical University of Warsaw, the study did not require the approval of the committee due to its questionnaire-based nature. The chairman of the bioethics committee was informed about the study and confirmed that it was not necessary to issue an approval. As the study was conducted online, ticking an informed consent statement was required to continue with the questionnaires. Completion of all measures by a participant was additionally equivalent to giving informed consent.

Statistical analyses were conducted using SPSS software version 27.0. A Wilcoxon signed-ranks test was used to answer whether there were any differences between the satisfaction with work-related aspects retrospectively reported before and during the pandemic. The Mann–Whitney U test and Chi-squared test were used to investigate group differences in factors associated with suicidal ideation. Spearman’s correlations and regression analyses were conducted to describe the association of satisfaction with work-related aspects during the pandemic with mental health variables. A multiple regression analysis was used to test if satisfaction with specific aspects of work safety significantly predicted the participants’ wellbeing and mental health.

## 3. Results

We obtained data from 367 healthcare professionals: 68 (18.5%) nurses and midwives, 29 (7.9%) paramedics, 85 (23.2%) physicians, 32 (8.7%) pharmacists and pharmacy technicians, 88 (23.9%) laboratory diagnosticians and 65 (17.7%) psychologists and therapists, who did not work remotely and were not quarantined at the surveyed time. There were 324 (88.3%) women and 43 (11.7%) men in the sample. The mean age was 37.16 ± 8.88 years. The mean work experience was 11.51 ± 9.24 years. A total of 219 (59.7%) people worked in the public sector, 106 (28.9%) in the private sector and 42 (11.4%) workers reported equal time between the sectors. In the event of mandatory quarantine, 215 (58.6%) respondents reported being able to safely quarantine for themselves and their close ones, while 152 (41.4%) denied having such a possibility.

A Wilcoxon signed-ranks test showed that satisfaction with all aspects of work safety decreased during the COVID-19 pandemic. Satisfaction with the clarity of procedures decreased from a median of 77% to 48% (z = −13.35, *p* < 0.001), access to personal protective equipment from 77% to 49% (z = −10.10, *p* < 0.001), the flow of information from 72% to 53% (z = −10.39, *p* < 0.001), financial stability from 80% to 52% (z = −12.58, *p* < 0.001) and general security from 80% to 50% (z = −14.29, *p* < 0.001). The means and standard deviations of satisfaction with these aspects of work safety are presented in Table 1.

A total of 254 (69.2%) participants obtained scores indicating poor wellbeing on the WHO-5 scale, 185 (50.4%) participants scored within the moderate or severe range of the PHQ-9 scale of depression and 56 (15.3%) reported suicidal ideation. None of the following factors were found to be associated with reports of suicidal ideation in this sample: age (z = −1.841, *p* = 0.066), length of work experience (z = −0.717, *p* = 0.474), sex (X^2^ (1, N = 367) = 0.39, *p* = 0.843), occupational sector (X^2^ (2, N = 367) = 1.324, *p* = 0.516) and the possibility to safely undergo quarantine (X^2^ (1, N = 367) = 2.928, *p* = 0.087).

A total of 150 (40.9%) participants scored within the moderate or severe range of the GAD-7 scale of anxiety, 83 (22.6%) participants scored within the moderate or severe range of the ISI scale of insomnia and scores of 125 (34.1%) pointed to subthreshold insomnia (Table 2). The distribution of participants’ scores on mental health scales is presented in Table 2.

A multiple regression analysis was performed with wellbeing and mental health variables as outcomes and satisfaction with work aspects during the epidemic as predictors to determine if wellbeing and mental health could be predicted as a function of satisfaction with work aspects during the epidemic. The correlation coefficients between the predictor variables are presented in Table 3, and the correlation coefficients between the predictor and outcome variables are in Table 4.

The flow of information and financial stability explained 15.3% of the variance in WHO-5 scores (R^2^ = 0.153, F(4, 362) = 16.318, *p* < 0.001), 16.8% of the variance in PHQ-9 scores (R^2^ = 0.168, F(4, 362) = 18.331, *p* < 0.001) and 12.6% of the variance in ISI scores (R^2^ = 0.126, F(4, 362) = 13.046, *p* < 0.001). Satisfaction with clarity of procedures, the flow of information and financial stability explained 18.9% of the variance in GAD-7 scores (R^2^ = 0.189, F(4, 362) = 21.139, *p* < 0.001). Satisfaction with access to personal protective equipment was not a significant predictor in any of the regression models. The unstandardized coefficients of the predictors in the models are presented in Table 5.

## 4. Discussion

In the current study, healthcare professionals reported a deterioration of satisfaction with all safety-related work aspects in time of the pandemic. Two work-related aspects most often predicted mental health outcomes in the regression analyses—the flow of information and financial stability—but the percentage of variability in the outcome variables predicted by these aspects did not exceed 19%.

Our study followed a similar methodology to a study by Young et al. [23], which allows us to compare reported prevalence rates of mental disorders’ symptoms. We found a higher percentage of respondents whose scores indicated mild-to-severe symptoms of depression (scores of ≥5 on the PHQ-9; 78.2% vs. 47%) and anxiety (scores of ≥5 on the GAD-7; 74.4% vs. 63%). The percentages of people reporting symptoms of depression and anxiety were also higher in our study than in a study by Szwamel et al. [24] but not compared with a study by Dziedzic et al. [25]. Reports of moderate and severe symptoms of insomnia were higher in a study by Krupa et al. [26] than in this study.

Furthermore, in our sample, the percentage of respondents who reported suicidal ideation was much higher than in a study by Young et al. (15.3% versus 5% [23]). None of the factors we investigated were shown to be linked to the reports of suicidal ideation in this sample (age, sex, length of work experience, occupational sector, possibility to safely undergo quarantine). Other factors which should be taken into account when examining possible factors related to the incidence of suicidal ideation among healthcare workers during epidemics should include burnout, post-traumatic stress disorder [27], support at work or perceived discrimination due to working in healthcare during the pandemic [28].

Other studies conducted during the COVID-19 pandemic have shown that access to PPE predicted better health and job satisfaction, less distress [29] and less anxiety [13]. Maciaszek et al. highlighted how appropriate protection was the main factor affecting mental health in their study [14]. However, the results of this research show that, when taking into account other work-related safety aspects, access to PPE is no longer a significant predictor of the mental health of healthcare workers. Our results are consistent with research from the SARS-CoV-1 outbreak, which emphasized that clear communication, the flow of information and collaboration were essential aspects of dealing with the epidemic at a hospital [2].

Apart from the flow of information, we also found that financial stability significantly predicts the mental wellbeing of the participants. In our survey, we included a question about satisfaction with financial stability. Financial aspects are crucial components of wellbeing. To the best of our knowledge, only one other study considered this aspect in research performed on the mental health of healthcare workers during the pandemic. A study by Nowrouzi-Kia et al. [30] showed that concerns about changing financial situations and income are associated with work performance and mental health among healthcare workers.

In Poland, most healthcare workers have more than one workplace (e.g., physicians work on average in 2.6 different places, according to data from Polish Supreme Medical Chamber [31]). During the pandemic, there was a regulation from the Health Ministry of Poland restricting healthcare professionals to working in only one main workplace, which forced healthcare workers to resign from additional workplaces temporarily and which caused a significant reduction in income. Being forbidden to work in more than one place together with risking an infection put a double strain on the workers. There are different types of employment contracts in Polish healthcare, and many of the workers are not eligible for sick leave. Therefore, if they do not have insurance against loss of income in case of health problems (which is not popular in Poland), they are left with no income in the case of SARS-CoV-2 infection. However, in this study, we did not survey the participants on whether they had more than one workplace before the pandemic and had to limit them to the one they were at when the outbreak happened, so we cannot link it directly to the mental health of this sample.

The main findings of our study show that satisfaction with the flow of information at work and financial stability are predictors of the mental health of medical personnel during a pandemic, adding crucial information to previous research on occupational factors playing a role in maintaining the mental health of essential workers. However, the regression models explained less than 20% of the variability in the mental health variables. It is possible that greater variability would be explained if factors such as support from peers and employers, the intensity of tasks and work schedule were included in the analyses [32].

In this study, satisfaction with work-related security was retrospectively assessed, which is not free from individual and situational bias. In addition, participants were self-selected, which may have led to a response bias from individuals more concerned with their workplace safety and mental health during the pandemic than if we were to sample responses from randomly selected healthcare professionals.

Moreover, it was impossible to differentiate whether the mental health symptoms were associated with specific work-related conditions during the pandemic or with the occupation itself. There was no comparator group outside of healthcare. Due to the sudden outbreak of the pandemic, we were not able to assess the study variables longitudinally instead of retrospectively. We were not able to account for mental health before the pandemic. Therefore, we associated current mental health with occupational satisfaction solely during the pandemic.

This study was observational, which does not allow for inferring causality. It lacks longitudinal follow-up. The methodology used in this study relies on self-report; however, using popular scales permits comparisons with other studies and among various groups of participants. The research was conducted solely in Poland, which prohibits any generalization of the results. The organization of healthcare work differs between countries. Future studies should include cross-cultural research.

## 5. Conclusions

This study showed that healthcare professionals reported a deterioration of satisfaction with all safety-related work aspects: clarity of procedures, access to personal protective equipment, the flow of information, financial stability and the general sense of security. Among these five investigated aspects, satisfaction with the flow of information at work and financial stability were found as significant predictors of the mental health of medical personnel during the first wave of the COVID-19 pandemic. This adds crucial information to previous research on occupational factors playing a role in maintaining the mental health of essential workers. Improving the flow of information is a cost-free strategy to better conditions at work for healthcare providers in times of crisis.

## Figures and Tables

**Table 1 ijerph-20-02870-t001:** Means and standard deviations of satisfaction with aspects of work (0–100%).

	Before	During	Wilcoxon	Difference
Clarity of procedures [%]	71.03 ± 23.35	45.54 ± 27.01	−13.35 ***	25.49 ± 28.32
Access to personal protective equipment [%]	69.86 ± 28.69	49.54 ± 31.65	−10.10 ***	20.32 ± 34.75
Flow of information [%]	66.68 ± 26.67	53.26 ± 30.67	−10.39 ***	13.43 ± 23.49
Financial stability [%]	71.91 ± 26.58	52.40 ± 30.10	−12.58 ***	19.52 ± 25.99
General security [%]	73.57 ± 24.55	48.40 ± 27.67	−14.29 ***	25.18 ± 25.50

*** *p* < 0.001.

**Table 2 ijerph-20-02870-t002:** Distribution of participants’ scores on the mental health scales.

	N	%
WHO-5 [points]: Wellbeing		
≤12 (<50%) Poor	254	69.2
≥13 (>50%) Good	113	30.8
PHQ-9 [points]: Depressive symptoms		
0–4: None	80	21.8
5–9: Mild	102	27.8
10–14: Moderate	79	21.5
15+: Severe	106	28.9
Suicidal ideation: Yes	56	15.3
GAD-7 [points]: Anxiety		
0–4: None	94	25.6
5–9: Mild	123	33.5
10–14: Moderate	91	24.8
15+: Severe	59	16.1
ISI [points]: Insomnia		
0–7: Not clinically significant	159	43.3
8–14: Subthreshold	125	34.1
15–21: Moderate	69	18.8
22–28: Severe	14	3.8

Note: GAD-7—General Anxiety Disorder-7; ISI—Insomnia Severity Index; PHQ-9—The Patient Health Questionnaire-9; and WHO-5—The World Health Organization-Five Well-Being Index.

**Table 3 ijerph-20-02870-t003:** Correlations between levels of satisfaction with aspects of work safety during the COVID-19 pandemic.

Satisfaction during the COVID-19 Pandemic				
	1	2	3	4
1. Clarity of procedures	-			
2. Access to personal protective equipment	0.585 ***			
3. Flow of information	0.518 ***	0.516 ***		
4. Financial stability	0.333 ***	0.376 ***	0.386 ***	
5. General security	0.474 ***	0.520 ***	0.518 ***	0.666 ***

*** *p* < 0.001.

**Table 4 ijerph-20-02870-t004:** Correlations between the outcome and predictor variables. Significantly correlated variables were entered into regression models.

		Satisfaction during the COVID-19 Pandemic				
	Age	Clarity of Procedures	Access to Personal Protective Equipment	Flow of Information	Financial Stability	General Security
WHO-5	0.048	0.266 ***	0.196 ***	0.320 ***	0.320 ***	0.430 ***
PHQ-9	−0.057	−0.218 ***	−0.195 ***	−0.333 ***	−0.351 ***	−0.492 ***
GAD-7	−0.103 *	−0.296 ***	−0.262 ***	−0.337 ***	−0.380 ***	−0.512 ***
ISI	0.017	−0.177 ***	−0.181 ***	−0.263 ***	−0.332 ***	−0.364 ***

Note: GAD-7—General Anxiety Disorder-7; ISI—Insomnia Severity Index; PHQ-9—The Patient Health Questionnaire-9; and WHO-5—The World Health Organization-Five Well-Being Index. NS – not significant; * *p* < 0.05; *** *p* < 0.001

**Table 5 ijerph-20-02870-t005:** Results of regression analyses.

	WHO-5					PHQ-9					GAD-7					ISI				
	B	SE	t	CI 95%		B	SE	t	CI 95%		B	SE	t	CI 95%		B	SE	t	CI 95%	
Clarity of procedures	*NS*					*NS*					−0.026 *	0.013	−2.074	−0.051	−0.001	−				
Access to personal protective equipment	Not a significant predictor in any of the models																			
Flow of information	0.035 ***	0.010	3.598	0.016	0.054	−0.051 ***	0.013	−4.039	−0.076	−0.026	−0.029 **	0.011	−2.692	−0.050	−0.008	−0.029 *	0.013	−2.286	−0.054	−0.004
Financial stability	0.034 ***	0.009	3.933	0.017	0.051	−0.054 ***	0.011	−4.772	−0.077	−0.032	−0.049 ***	0.010	−5.144	−0.068	−0.031	−0.057 ***	0.011	−4.968	−0.079	−0.034
*F*	16.318 ***					18.331 ***					21.139 ***					13.046 ***				
R^2^	0.153					0.168					0.189					0.126				

Note: GAD-7—General Anxiety Disorder-7; ISI—Insomnia Severity Index; PHQ-9—The Patient Health Questionnaire-9; and WHO-5—The World Health Organization-Five Well-Being Index. NS—not significant; * *p* < 0.05; ** *p* < 0.01; and *** *p* < 0.001.

## Data Availability

The data that support the findings of this study are available from the corresponding author, K.G., upon reasonable request.

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
