# Peer review of "Which Aspects of Work Safety Satisfaction Are Important to Mental Health of Healthcare Workers during COVID-19 Pandemic in Poland?"

_ijerph, 2023, doi:10.3390/ijerph20042870_

Round 1

Reviewer 1 Report

The present paper aimed at investigating the impact of COVID-19 on mental health symptoms and on wellbeing and work safety satisfaction of Polish healthcare workers in May 2020. Three hundred sixty-seven subjects completed an online survey including work satisfaction during the pandemic and retrospectively before the outbreak, besides a brief psychiatric assessment. The results of the present online survey showed a decrease of healthcare workers’ satisfaction during the COVID-19 pandemic due to several factors, including unclear procedures, lack of PPE and information, besides financial problems. Some of these variables seemed to predict an impairment in the psychiatric assessment scores.

 Comments:

1)          In the Introduction section, more detailed information about the impact of COVID-19 on the Polish population and Polish healthcare workers/system during the present survey (May 2020) should be provided.

2)          Introduction,   Lines 38-52. The literature review is scarce and can be improved for this background (see e.g. 10.1016/j.jad.2021.10.128; 10.1016/j.psychres.2021.113763; 10.1016/j.ijnurstu.2021.104002; 10.3390/ijerph18041435; 10.1016/j.idh.2021.12.002; 10.2147/PRBM.S330676; 10.3390/healthcare10071330; 10.2174/1745017902117010242 about the impact of the COVID-19 pandemic “first wave” on mental health outcome in healthcare workers).

3)          As reported in Methods and Results, subjects were recruited from physicians, nurses, paramedics, pharmacist, psychologist, laboratory personnel, etc. There were any differences on the study variables across the different healthcare profiles? It would be of interest to Readers if the Authors provided data on this point.

4)          Interestingly, Authors reported that about 15% of subjects showed suicidal ideation by means of PHQ-9. This is an interesting point. How can this high percentage be discussed? Is it possible to evaluate which are the predictor variables of suicidal ideation? Are there healthcare profiles or other socio-demographic (e.g. age, gender, etc.) or clinical (e.g. the pandemic-related post-traumatic stress) variables that may affect this result?

5)          Table 5: please report also B(SE), T and CI 95% values of each multiple regression analyses.

6)          Discussion, lines 143-146: The regression analyses’ models explained a percentage of variability in the outcome variables predicted lower than 20%. This is an important limitation of the model proposed. How do the Authors discuss this point? Probably, there other variables that could affect the scores on the psychiatric and wellbeing assessments (cfr. point 4).

Reviewer 2 Report

Thank you for the opportunity to review this study which explored the aspects of work safety satisfaction relevant to the mental health and well-being of healthcare workers during the COVID-19 pandemic in Poland.  

Major comments

The well-being and mental health of healthcare workers during the COVID-19 pandemic is a subject of great interest, which has been addressed in numerous studies carried out in different countries of the world, including Poland. Therefore, the authors must emphasize the novelty and originality of their study and how their results are relevant for the healthcare system in Poland. 

Introduction

The authors must describe the problematic aspects faced by healthcare workers in Poland during the COVID-19 pandemic and which justified the realization of this study.

Materials and Methods

A description of the instruments used for data collection is necessary. 

What statistical tool was used for data analysis?

Results

A more detailed description of the group participants is necessary- for instance, how many persons from each professional category participated, the workplace of the participants. The workplace of the participants might be relevant as healthcare providers working in clinics and especially in the ICU, Infectious Diseases or ER were more exposed to the infected patients and faced more acutely the problematic issues of the health systems generated by the COVID-19 pandemic compared to the healthcare providers working in the pharmacies or laboratories. 

The participants in this study belong to very diverse professional categories, which could be a bias for the results obtained.

Discussion

The discussion should be expanded, to analyze all the significant results and relate them to other similar studies in the literature and particularly to the ones conducted in Poland.

Lines 169-170- “In Poland, most healthcare workers have more than one workplace”- is this the case for all healthcare workers categories? Participants in this study belong to quite different professional categories. Is it correct to assume that all categories of healthcare providers work in more than one place and were negatively affected by the interdiction of working in more than one place? 

Minor comments

- Line 18- What do the authors understand by "general security"?

- Lines 74, 219- please replace “consent” with “approval”

- It is not clear whether data were collected anonymously or not.

- Lines 140-141- it is not clear what the authors mean by “This study answered the call for research to identify predictors of mental health of healthcare providers during the COVID-19 pandemic”. What call are the authors talking about?

- Lines 166-168- In which sense “concerns about changing financial situation and income are associated with work performance and mental health among healthcare workers”?

Round 2

Reviewer 1 Report

Thank you to the Authors for addressing my suggestions. I have no further comments to raise.

Reviewer 2 Report

I congratulate the authors for the revised version of their manuscript that address all my comments and suggestions.